

# Environmental DNA and visual encounter surveys for amphibian biomonitoring in aquatic environments of the Ecuadorian Amazon

Walter Quilumbaquin[1], Andrea Carrera-Gonzalez[1,2],
Christine Van der heyden[3] and H. Mauricio Ortega-Andrade[1]

[1] Biogeography and Spatial Ecology Research Group, Universidad Regional Amazónica Ikiam, Tena, Napo, Ecuador
[2] Molecular Biology and Biochemistry Lab, Universidad Regional Amazónica Ikiam, Tena, Napo, Ecuador
[3] Health and Water Technology Research Centre, Department of Biosciences and Industrial Technology, HOGENT–Univesity of Applied Sciences and arts, Gent, Belgium

Corresponding author
Walter Quilumbaquin,
arman7085@gmail.com

## ABSTRACT

**Background:** The development of anthropogenic activities has generated a decline in aquatic fauna populations, and amphibians have been the most affected. The decline of batrachofauna is concerning, as 41% of all species worldwide are endangered. For this reason, rapid, efficient, and non-invasive biodiversity monitoring techniques are needed, and environmental DNA (eDNA) is one such tool that has been sparsely applied in Ecuador. This technique has allowed scientists generates information on species diversity and amphibian community composition from a water sample. This study applied eDNA-based biomonitoring analyses and visual encounter surveys (VES) as inventory techniques to identify the diversity of aquatic amphibians in the Tena River micro-basin (TRMB).

**Methods:** The experimental design was divided into three components: (1) fieldwork: all amphibians were recorded by the VES technique and water samples were collected; (2) laboratory work: DNA isolation from amphibian tissue samples and eDNA-containing filters, amplification, electrophoresis, and sequencing were performed; (3) Data analysis: a local DNA reference database was constructed, and eDNA sequence data were processed for classification, taxonomic assignment, and ecological interpretation.

**Results:** Using both eDNA and VES, we detected 33 amphibian species (13 with eDNA only, five with VES only, and 15 with both methods). These species belonged to six amphibian families: Hylidae being the richest with 14 species (three eDNA, one VES, and 10 with both methods), followed by Strabomantidae with nine species (six eDNA, one VES, and two with both methods). All families were detected with both methods, except for the Aromobatidae, having one single record (*Allobates* aff. *insperatus*) by VES. Individually, eDNA detected 28 species and had a detection probability (DP) of 0.42 CI [0.40–0.45], while VES recorded 20 species with a DP of 0.17 CI [0.14–0.20]. Similarly, using VES, *Cochranella resplendens* was detected for the first time in TRMB, while with eDNA, four mountain frogs *Pristimantis acerus*, *Pristimantis eriphus*, *Pristimantis mallii*, and *Pristimantis* sp. (INABIO 15591) previously recorded at 1,518 m.a.s.l. at altitudes below 600 m.a.s.l. were detected.

**Conclusions:** Results obtained in this study showed that eDNA-based detection had a greater capacity to detect amphibians in aquatic environments compared to VES. The combination of VES and eDNA improves the sensitivity of species detection and provides more reliable, robust, and detailed information. The latter is essential for developing conservation strategies in the Ecuadorian Amazon.

# INTRODUCTION

Aquatic ecosystems provide ecosystem services, such as local climate regulation, carbon capture, nutrient cycling, food production, and more (*Celi & Villamarín, 2020*; *Céréghino et al., 2014*). They harbor populations of diverse taxonomic groups that form complex ecological networks threatened by anthropogenic activities and climate change (*Cushing & Allan, 2009*; *Hassall, 2014*). The case of amphibians is alarming from an ecological and conservation perspective because species that were once frequently observed are now uncommon and endangered (*Ortega-Andrade et al., 2021*). Moreover, several of them still need to be described taxonomically (*Pereira, 2014*), indicating the urgent need to study them to understand their role and importance in the environment to determine conservation and/or restoration actions and their potential biomedical applications (*Ortega-Andrade et al., 2021*; *Stuart et al., 2004*; *Ron, Merino & Ortiz, 2019*).

Traditional metrics for monitoring species diversity and amphibian community composition are based on data generated by classical techniques such as visual encounter surveys (VES), traps, and bioacoustic recordings (*Heyer et al., 2001*; *Aguirre León, 2009*; *Bosch & García Padrón, 2017*). Applying these techniques requires expertise in the identification, multiple visits to the study sites, and many working hours. Moreover, regions with high biodiversity experience a reduction in detection sensitivity, especially when dealing with species with low population density (*Heyer et al., 2001*; *Brozio et al., 2017*). These factors complicate the efficient development of amphibian biomonitoring, detection, and conservation strategies (*Brozio et al., 2017*; *Barata, Griffiths & Ridout, 2017*; *Lopes et al., 2017*). Environmental DNA (eDNA)-based techniques can be implemented to address these problems. Due to its simplicity and power, eDNA has generated much interest in biodiversity monitoring programs as an alternative or complement to traditional inventory methods (*Barata, Griffiths & Ridout, 2017*; *Bohmann et al., 2014*; *Taberlet et al., 2018*).

The eDNA technique involves the capture and analysis of cells and traces of free DNA present in the environment ("eDNA") for the non-invasive detection and monitoring of various organisms (*Bohmann et al., 2014*; *Pedersen et al., 2015*). All living organisms, independent of their size, shape, or ecological niche, release DNA molecules into the environment intracellularly or extracellularly through feces, gametes, skin cells, *etc.*

(*Taberlet et al., 2018*; *Roh et al., 2006*; *Harper et al., 2019*). Environmental samples from soil, water, or air thus contain valuable information about ecosystem compositions and dynamics. Due to recent advances in sample preparation (new kits and laboratory devices) and the development of more affordable next-generation sequencing technologies, such as MinION from Oxford Nanopore Technologies (ONT), genetic analysis is more reasonable and accessible than ever before (*Harper et al., 2019*; *Maestri et al., 2019*).

Worldwide, the efficiency of the eDNA-based technique has been evaluated for monitoring invasive species (*Van der heyden et al., 2021*; *Riascos et al., 2018*), endangered species (*Brozio et al., 2017*), and species with low population density (*Lopes et al., 2017*). In aquatic environments, the use of the eDNA method has focused on the biomonitoring of fishes (*Shaw et al., 2016*; *Evans et al., 2017*) and amphibians (*Lopes et al., 2017*; *Ficetola, Manenti & Taberlet, 2019*), as these organisms are relatively easier to detect because they can produce and release more DNA into the environment than others (*Taberlet et al., 2018*; *Harper et al., 2019*; *Valentini et al., 2016*). The application of eDNA for monitoring amphibians in aquatic environments is constantly growing (*Harper et al., 2019*; *Wang et al., 2021*; *Sasso et al., 2017*). This is due to the accessibility and availability of the sample (water), its applicability in different types of water bodies (*i.e.*, pond, lake, river, or sea), and its potential to identify species individually or the abundance of several species in complex ecosystems (*Taberlet et al., 2018*; *Valentini et al., 2016*; *Baetens, 2019*; *Takahashi et al., 2020*). These characteristics make it an ideal technique for biomonitoring in areas rich in water sources and high amphibian biodiversity, such as the Ecuadorian Amazon.

The Tena River basin, located in the province of Napo in the Ecuadorian Amazon, is home to a wealth of water resources. Its two main rivers are the Tena (forming the Tena River micro-basin, TRMB) and the Pano. Both have tributaries that cover a drainage area of about 235 km$^2$ (*Hurtado-Pidal et al., 2020*). The TRMB drains water from diverse ecosystems ranging from Andean páramo to tropical rainforests from 500 to 2,500 m.a.s.l. (*Gobierno Provincias de Napo, 2015*). The diversity of habitats in this micro-basin correlated with rich amphibian biodiversity (*Ron, Merino & Ortiz, 2019*; *Lessmann et al., 2016*).

Ecuador is one of the most biodiverse countries in the world, and the third in amphibian diversity, with a total of 669 formally described species, of which 46.6% are endemic (*Ortega-Andrade et al., 2021*; *Ron, Merino & Ortiz, 2019*). Napo harbors the highest species richness among the Amazonian provinces, with 210 species recorded (*Ron, Merino & Ortiz, 2019*; *Ordóñez, Valle & Veintimilla, 2011*). In the TRMB, which includes the Colonso Chalupas reserve, 50 species have been reported (*Ron, Merino & Ortiz, 2019*; *Ordóñez, Valle & Veintimilla, 2011*; *Coloma & Tapia, 2015*). At a more local level, approximately 25 amphibian species have been recorded at the Universidad Regional Amazónica Ikiam (Ikiam) campus and surrounding communities (*Ron, Merino & Ortiz, 2019*; *Ordóñez, Valle & Veintimilla, 2011*; *Coloma & Tapia, 2015*).

In South America, amphibian biomonitoring programs with eDNA have been implemented in the Bolivian and Brazilian Amazon (*Sasso et al., 2017*; *Bálint et al., 2018*). In Ecuador, amphibian biomonitoring studies with eDNA are limited. Most notably, it has been used for monitoring marine ichthyofauna and artisanal and industrial fishing vessels
(*Willette et al., 2021*). In the Ecuadorian Amazon, the use of eDNA is beginning to gain interest through research projects developed at Ikiam (*Van der heyden et al., 2021*). So far, amphibian monitoring in TRMB has consisted of capture, VES, and audio recordings (*Heyer et al., 2001*; *Ortega-Andrade, 2010*; *Tapia Del Águila, López-Rojas & Pérez-Peña, 2020*). However, this study sets a precedent in implementing eDNA to improve amphibian monitoring in the TRMB.

This is the first study in the Ecuadorian Amazon using eDNA to identify amphibians associated with water bodies. The main research objectives were: (1) to identify aquatic amphibian diversity in the TRMB using eDNA and VES; (2) to compare the abilities of eDNA-monitoring and VES to detect amphibian species in aquatic environments; (3) to develop a scientific collection and a local reference taxonomic and genetic database of amphibians in the TRMB.

## MATERIALS AND METHODS

### Study site

The study site is within the TRMB (Napo Province, Ecuador), in the Amazon Piedmont, 8 km west of the city of Tena, Atacapi community (0°56′51°S; 77°51′37°W). It has an altitudinal range between 600 to 710 m.a.s.l. and an annual precipitation of 3,500 mm. Adjacent to the Ikiam, it hosts numerous water bodies, including rivers, streams, and ponds (*Ministerio del Ambiente del Ecuador, 2013*). In the sampling sites, we collected specimens and eDNA samples from three sections of the Tena River (TRP1–TRP3) and three ponds (ChP1–ChP3) with permanent water (*i.e.*, those that have water throughout the year) adjacent to the Tena River (Fig. 1).

### Permission

This research project was carried out through the Framework Agreement for Access to Genetic Resources No. MAATE-DBI-CM-2021-0177 issued by the Ministerio de Ambiente, Agua y Transición Ecológica and within the project entitled: Detection of keystone species of fauna (threatened, invasive) and microorganisms (pathogens and microbiota) associated with aquatic ecosystems of Ecuador, with molecular techniques of environmental DNA and metabarcoding.

### Sterilization process

Before water sampling, polyethylene bottles, forceps, Petri dishes, and filtration systems were sterilized at the laboratory. For this process, the material was soaked in 50% (v/v) or 3% (m/v) bleach (6% Clorox, active ingredient: sodium hypochlorite, Oakland, CA, USA) for 3 min. Next, the bleach residues were removed with distilled water (1 min), then with 70% ethanol for 3 min. Finally, the materials were autoclaved at 1.02 atm and 121 °C for 60 min (*Baetens, 2019*). This sterilization process was performed for each sampling point to remove DNA residue and avoid cross-contamination between collection points (*Riascos et al., 2018*; *Shaw et al., 2008*).

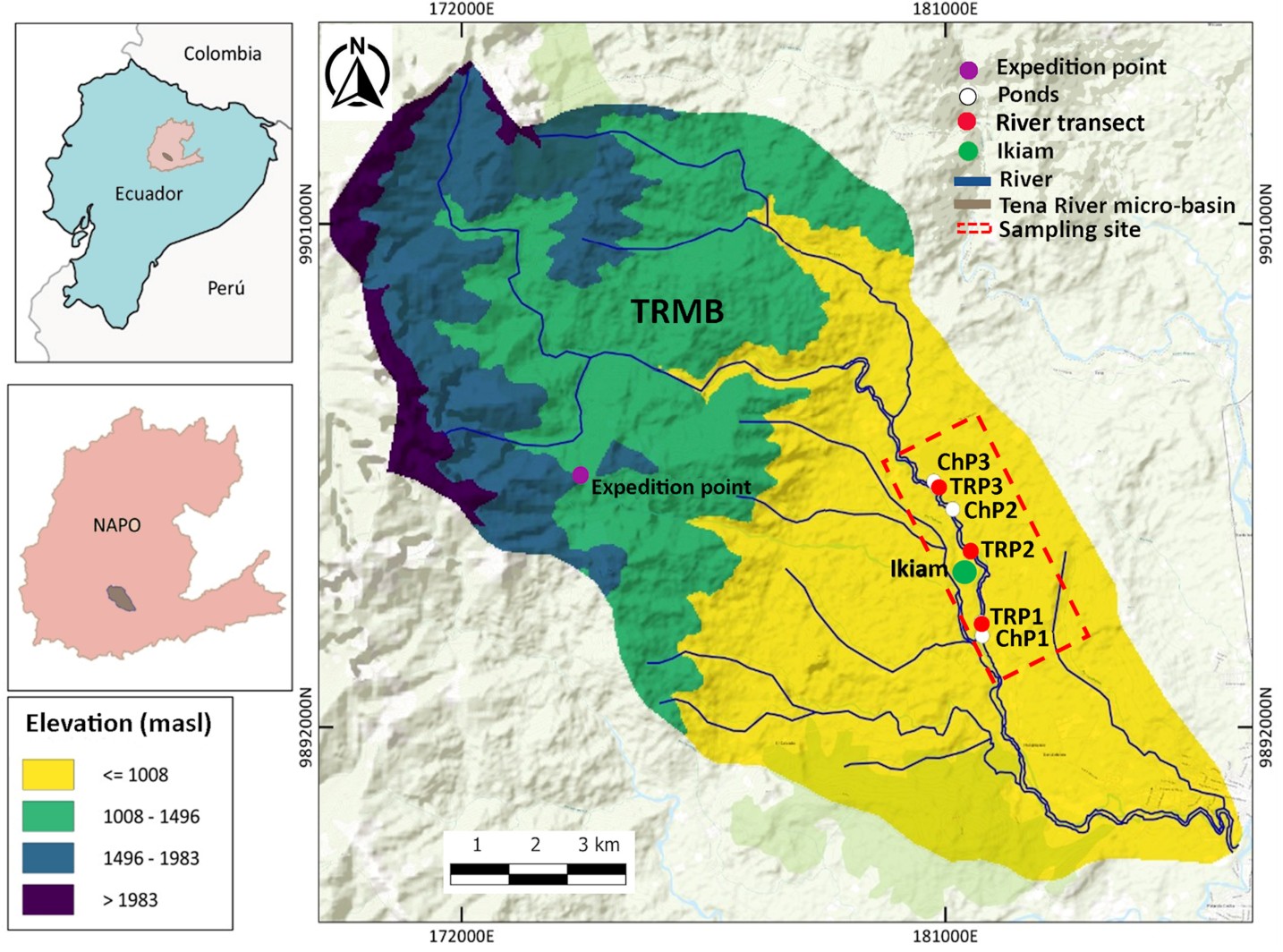

**Figure 1 Study area in the Tena River Micro-Basin (TRMB), Napo, Ecuador with the respective terrain elevation.** It consists of six sampling sites: three in ponds (ChP1, ChP2, ChP3, White) and three in river transects (TrP1, TrP2, TrP3, red). It has an exploration site (purple) in the Chalupas Colonso Reserve, where mountain amphibians were collected. Adjacent to the sampling area is the Universidad Regional Amazónica Ikiam (green). Detailed geographic location in Table S1 (Supplemental Material).

## Fieldwork

VES and water samples collection for eDNA were conducted simultaneously from January to March 2020 and February to April 2021. Both methods were performed once per month at six sites: three in ponds (ChP1–ChP3) and three in river transects (TRP1-TRP3) (Fig. 1; Table S1, Supplementary Material). Each site was surveyed six times in the wet season, specifically on days with little or no precipitation. The chronological order of sampling methods was as follows: (1) collection of water samples (morning) and (2) observation and recording of amphibians (evening). This order prevents the contamination of the environment, especially ponds, as VES is an invasive method that could influence eDNA results if performed earlier (*Heyer et al., 2001*).

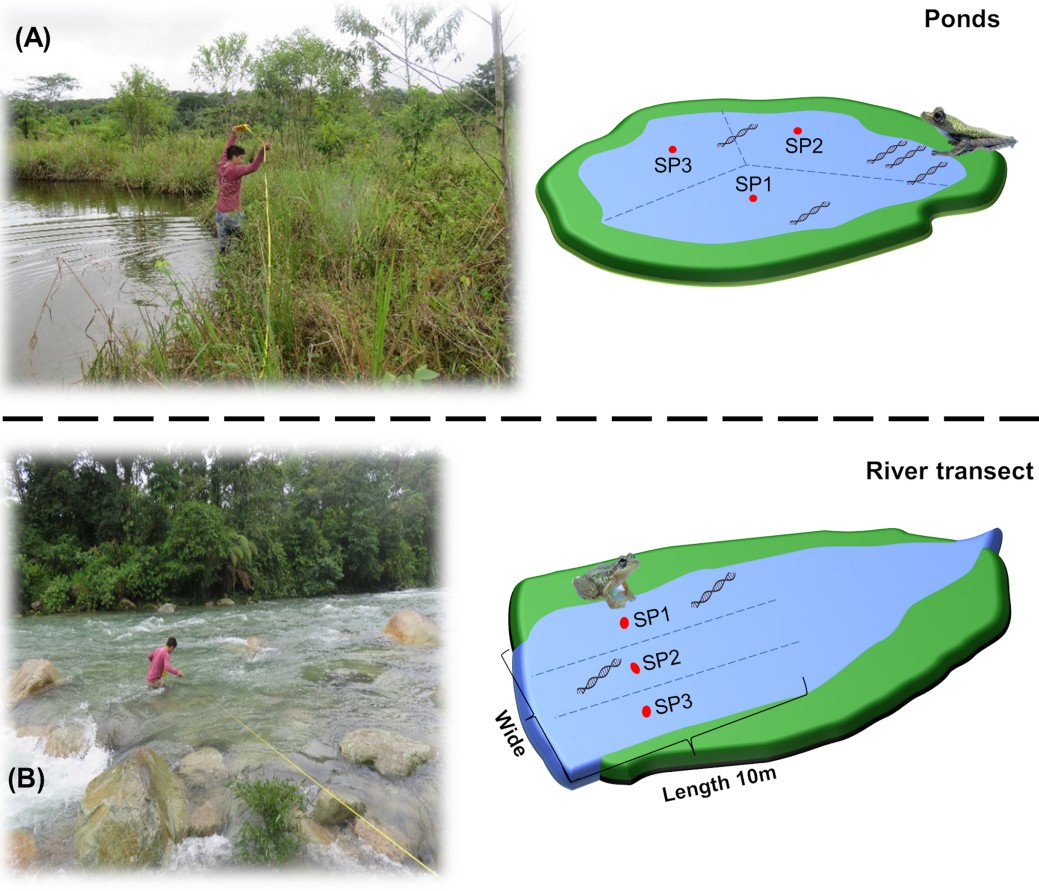

**Figure 2 Graphical representation of water collection points in Ponds and river transects.**
(A) measurement of the perimeter and division of the water collection points in the ponds.
(B) Measurement of the width and division of the water collection points in the river transects. SP1, SP2
and SP3 refers to the divisions performed at the sampling site prior to environmental sampling.

The collection of water samples was performed by two people in the morning (between 06:00 and 07:00 h. GMT-5). The perimeter of each site was measured and divided into three points (Fig. 2). Sterile 1 L polyethylene bottles were preconditioned by filling them with 50 ml of water from the site, shaking, and discarding away from the sampling site (*i.e.*, on the shore). This process was performed by three times to remove all remnants of bleach used in the sterilization process, which can disintegrate eDNA (*Shaw et al., 2008*). At each point, 1 L of water was collected, with a zigzag sweep 30 cm below the surface with a final volume for each sampling site of 3 L. To avoid cross-contamination in the field between sites, we used new nitrile gloves for each collection point, and the materials (*e.g.*, boots, plastic thermal box, among others) were sterilized with 10% (v/v) bleach for 1 min and 70% ethanol for 1 min, before moving to the next sampling point. At the same time, a bottle with 1 L of ultrapure water was added to the assay as a negative field control to monitor the contamination during transport and storage. The samples were transported in

plastic thermal boxes at 10 °C to prevent DNA degradation and bacterial growth (*Wang et al., 2021*).

The VES was performed by three researchers at night, between 20:00 and 00:00 h., for 30 min per site. Observation and identification of amphibians in the ponds were carried out through a slow (approx. 2 m/min) and random movement that covered the entire possible perimeter (approx. 20 m), checking vegetation, water body, and soil. For the river transects, the process was similar, but the observation was along the banks (approx. 25 m).

## Filtration of environmental samples

The water samples for eDNA and negative controls were filtered at the Ikiam immediately upon arrival at the lab (approx. 2 hrs.). The process was performed using a sterilized glass filtration system (Merck®, Rahway, NJ, USA) connected to a vacuum pump (GAST) and nitrocellulose membranes (PORAFIL® NC) with a pore size of 0.45 μm and 47 mm in diameter. Before filtration, the samples were homogenized from water collected at each collection point (Fig. 2), obtaining a final volume of three liters per sampling site. Samples were filtered until the filters were clogged and the water flow stopped. The volume of water used to clog a filter was 1 L for river samples and approximately 750 mL for pond samples. 1 L of ultrapure water was filtered as negative controls, and the same processes were followed for the environmental samples. Filters were passed to a sterile petri dish using sterilized forceps. We carefully cut filters into small pieces using sterile scalpels (No. 11), placed them in sterile 1.5 mL microtubes, and stored them at −20 °C. The samples collected in 2020 were processed seven months after collection, while those from 2021 were processed 1 week after the last collection.

## Processing and identification of collected specimens

Specimens were transported to the laboratory, photographed, and their morphological characteristics were described in life for taxonomic identification. All specimens were sacrificed using an anesthetic (2% Lidocaine) applied topically on the skin (*Angulo et al., 2006*). Samples of liver and leg muscle tissue were extracted from each specimen. Tissues were preserved in 2 mL microtubes with 96% ethanol and stored at −20°C until DNA extraction. Finally, each specimen received a unique code and was identified with dichotomous keys (*Ortega-Andrade, 2010*; *Angulo et al., 2006*). When dichotomous keys were absent, Bioweb Ecuador (https://bioweb.bio/faunaweb/amphibiaweb/) and the Ikiam amphibian collection were referenced. All specimens were preserved in the Laboratorio de Biología Integrativa Ikiam-Instituto Nacional de Biodiversidad (INABIO) scientific collection.

## Molecular assays for tissue samples

Genomic DNA extraction was performed using the Wizard® Genomic DNA Purification kit (Promega, Madison, WI, USA), with 5–10 mg of liver or muscle tissue, following the manufacturer's protocol (Supplemental Material). The NanoDrop™ One/Onec Microvolume UV-Vis Spectrophotometer (Thermo Scientific, Waltham, MA, USA) was used to determine the purity and quantity of genomic DNA. A 600 bp fragment of the 16S

ribosomal RNA gene (16S rRNA) was amplified with a miniPCR™ mini 16 thermocycler (MiniPCR, Cambridge, MA, USA). The final reaction volume was 25 μL and consisted of 2 U of Platinum Taq DNA Polymerase (Invitrogen, Carlsbad, CA, USA), nuclease-free water, 1X PCR Buffer, 1.5 mM MgCl$_2$, 0.2 mM dNTPs Mix, 0.2 μM of the markers 16 SA-L (5′-CGC CTG TTT ATC AAA AAC AT-3′), 0.2 μM of 16 Sb-H (5′-CCG GTC TGA ACT CAG ATC ACGT-3′) (*Vences et al., 2005*), 50 ng/μL bovine serum albumin (BSA; Invitrogen, Carlsbad, CA, USA) and 50 ng/uL of genomic DNA. The thermocycler program was as follows: 5 min at 95 °C, followed by 35 cycles of 30 s at 95 °C, 30 s at 59 °C and 45 s at 72 °C. A final extension was performed for 5 min at 72 °C and cooling to 4 °C. PCR products were visualized on 2% agarose gels with the BlueGel™ electrophoresis system (MiniPCR, Cambridge, MA, USA) using Diamond™ Nucleic Acid Dye stain (Promega, Madison, WI, USA).

Prior to sequencing, PCR products were purified using Exonuclease I and Shrimp Alkaline Phosphatase (New England BioLabs, Ipswich, MA, USA), following the manufacturer's guidelines. Sanger sequencing was performed by Macrogen (Seoul, South Korea). Data were aligned and edited using the free software Ugene V.41.0 (Unipro, Novosibirsk, Russia, http://ugene.net/) to generate a consensus sequence for each species. The sequences were compared with the National Center for Biotechnology Information (NCBI) database through the Basic Local Alignment Search Tool (BLAST, https://blast.ncbi.nlm.nih.gov/Blast.cgi) algorithm to assign the genus and species of every sample collected. A threshold of >98% pairwise identity was used to assign species.

## Molecular assays for environmental samples

The eDNA extraction was performed using the method described in *Baetens (2019)* with modifications (Supplemental Material). The DNA samples obtained were quantified using the Qubit 4 Fluorometer (Invitrogen, Carlsbad, CA, USA) and NanoDrop™ One/Onec Microvolume UV-Vis Spectrophotometer (Thermo Scientific, Waltham, USA). A short 300 bp fragment of the 16S rRNA gene was amplified with a miniPCR™ mini 16 thermal cycler (MiniPCR, Cambridge, MA, USA). The final reaction volume was 25 μL and contained 15-30 ng/μL extracted DNA, 1 U of Taq DNA Polymerase (Bio Basic, Markham, IL, USA), nuclease-free water, 1X Taq reaction Buffer, 2 mM MgSO$_4$, 0.2 mM dNTPs Mix, 0.2 μM of Vert 16S eDNA F1 markers (5′-TTT CTG TTG GTT GGT GCT GCT GAT ATT ATT GC AGA CGA GAA GAC AGA CCC YDT GGA GCTT-3′), 0.2 μM of Vert 16S eDNA R1 (5′-ACT TGC TCG CTC TCT ATC TTC GAT CCA ACA TCG AGG TCG TAA-3′) (*Vences et al., 2016*) and 50 ng/μL bovine serum albumin (BSA, Invitrogen, Carlsbad, CA, USA). The thermocycler program was as follows: 3 min at 94 °C, followed by 30 cycles of 30 s at 94 °C, 60 s at 58 °C and 90 s at 72 °C; continued with a final extension of 10 min at 72 °C and cooling to 4°C. PCR products were visualized on 2% agarose gels in a BlueGel™ electrophoresis system (MiniPCR, Cambridge, MA, USA) using Diamond™ Nucleic Acid Dye staining (Promega, Madison, WI, USA). After visualizing all 36 samples and negative controls, six samples were discarded because no band was present (Figs. S1 and S2).

Prior to sample indexing, the PCR products (30 samples and two negative controls) were purified using Exonuclease I and Shrimp Alkaline Phosphatase (New England BioLabs, Ipswich, MA, USA), following the manufacturer's guidelines. An indexing barcode was assigned to each sample using the Barcoding Expansion Pack 1–96 PCR kit (EXP-PBC096; ONT, Oxford, UK). The final reaction volume was 50 μL, containing 1 μL of PCR Barcode (BC01–BC32), 24 μL of PCR product, and 25 μL of LongAmp Taq 2X Master Mix (New England BioLabs, Ipswich, MA, USA). The thermal profile was as follows: 3 min at 95 °C, followed by 15 cycles of 15 s at 95 °C, 15 s at 62 °C and 30 s at 65 °C, a final extension of 10 min at 65 °C and cooling to 4 °C. 10 μL of each barcoded product was pooled to create the initial library.

The initial library was purified with Agencourt 1.8x AMPure XP beads (Beckman Coulter, Brea, CA, USA) and quantified with the Qubit 4 Fluorometer kit (Invitrogen, Carlsbad, CA, USA). The pooled library was diluted to 1 μg in 47 μL nuclease-free water. End-prep and adapter ligation were performed according to the barcode PCR protocol (96) SQK-LSK109 amplicons (ONT, Oxford, UK), using the NEBNext FFPE DNA repair mix (M6630), NEBNext Ultra II End repair/dA-tailing module (E7546) and ligation sequencing kit 1D (SQK-LSK109; ONT, Oxford, UK). A MinION flow cell was used for sequencing (FLO-MIN106D; ONT, Oxford, UK). Prior to library loading, the flow cell was prepared using the flow cell priming kit (EXP-FLP002). The loading mix had a final volume of 75 μL and was prepared with 7.5 μL sequencing buffer (SQB), 25.5 μL loading beads (LB), and 35 fmol of DNA library in 12 μL.

Sequencing was performed on a MinION mk1b (ONT, Oxford, UK). The device used MinKNOW v21.06.0 which has integrated Guppy v5.0.11 software to control the sequencer, collect sequences in real-time, filter reads by quality >8, and perform Basecalling and demultiplex reads by barcode. The sequencing time was 15 h. After the sequencing time elapsed, the equipment was disconnected and the flow cell (FLO MIN106 R9.4.1; ONT, Oxford, UK) was cleaned with the flow cell cleaning kit (EXP-WSH004; ONT, Oxford, UK) following the factory instructions, and the cell was stored at 4 °C.

## 16S rRNA gene reference database

Based on field sampling, Ikiam amphibian reference collection, and literature reports on the diversity and distribution of amphibian species in the TRMB (*Ron, Merino & Ortiz, 2019*; *Ordóñez, Valle & Veintimilla, 2011*; *Coloma & Tapia, 2015*), we defined a list of 62 amphibian species that could be detected by eDNA (Table S2, Supplemental Material). With this list, a 16S rRNA gene reference database was compiled from three sources of information: (1) tissues of 43 specimens collected in this study, corresponding to 20 species (Table S3, Supplemental Material); (2) sequences generated by other research projects at the Universidad Regional Amazónica Ikiam; (3) sequences deposited in the database of the National Center Biotechnology Information (NCBI).

The 16S rRNA gene sequences of 59 of the 62 species cited in the TRMB were compiled. The three species without reference sequences were *Hyloscirtus albopunctulatus, Boana appendiculata*, and *Caecilia dunni*. Furthermore, all Ecuadorian anuran 16S rRNA sequences available in NCBI were added (https://bioweb.bio/). Since the TRMB is mainly

unexplored, these sequences were included for more taxonomic coverage. In addition, sequences from mammals, fish (*Barriga, 2012*), birds (*Rios et al., 2018*), reptiles (*Ron, Merino & Ortiz, 2019*) (common to the sampling site), and bacteria (*Dowden et al., 2020*) (sequenced with the ONT) were added. Since the eDNA primers are vertebrate-specific, these references help sort non-anuran eDNA reads to their closest taxonomic groups (Table S4, Supplemental Material).

Finally, the reference database was cleaned: verifying the taxonomic description at the phylum, class, order, family, genus, and species level of each sequence, removing redundant sequences, and checking that each sequence had a unique identifier. To evaluate the functionality of the eDNA primers, In-silico PCR was performed using ecoPCR (*Ficetola et al., 2010*), with a maximum of two mismatches per primer binding site. The minimum and maximum length of the generated amplicon (excluding primers) was 250 and 500 bp, respectively.

## Bioinformatics and statistical analysis

Raw fast5 files were basecalled with the 'Fast' algorithm in Guppy v5.0.11. basecalled Fastq files were processed with Python v3.10.1. and a docker image in Docker Desktop 4.1 (https://docs.docker.com/). The analysis was performed independently for each sequenced sample. NanoFilt v2.5.0 package (https://github.com/wdecoster/nanofilt) was used to filter reads based on quality scores (Q > 12) and remove all sequences below 200 nucleotides. Sequences of the adapters and primers were found and removed using the Cutadapt v4.1 tool (https://github.com/marcelm/cutadapt). Amplicon_sorter (https://github.com/avierstr/amplicon_sorter) was used to cluster and sort sequences according to similarity and length and to build a robust consensus sequence (reduces sequencing errors (*Dowden et al., 2020*)) for each group present in the eDNA data. This tool was run following the steps described in *Vierstraete & Braeckman (2022)*, setting a size range between 200 and 500 bp (script, commands, and read files for each barcode and sequenced controls are available from https://doi.org/10.6084/m9.figshare.c.6321395.v1).

Taxonomic assignment of consensus sequences obtained from eDNA (Molecular Operational Taxonomic Units—MOTUs) was performed using two bioinformatic tools: (1) BLAST+ v2.13.0 (https://anaconda.org/bioconda/blast) and (2) the assignTaxonomy function of DADA2 v2 1.18, (https://benjjneb.github.io/dada2/assign.html). BLAST+ was used to search for matches between the reference sequence database and MOTUs. Percent identity, alignment length, e-value, and bit score were considered. For species-level assignment of amphibian MOTUs, we applied an alignment length of approx. 300 bp and a sequence similarity threshold >97% (*Lopes et al., 2017*; *Bonin, Guerrieri & Ficetola, 2022*; *Gwak & Rho, 2020*), and a genus level threshold of 90–96%. MOTUs from the other vertebrate groups were identified using a 90–99% similarity threshold and separated from amphibian MOTUs without further analysis. MOTUs showing <90% similarity to the reference databases were removed. The *assignTaxonomy* function (DADA2) generated a complete taxonomic assignment from domain to species and served to corroborate the results obtained with BLAST+.
The Detection probability (DP) was estimated using a single-species, single-season occupancy model (*Schultz & Lance, 2015*; *Furlan et al., 2016*) with the UNMARKED package in R.4.1.2 (*R Core Team, 2021*; *Fiske & Chandler, 2011*). This standard model was fitted with presence-absence matrices (PAMs) generated for each sampling method (eDNA and VES). PAMs were constructed for each species recorded by both methods during the six months of sampling at the six collection sites. We considered presence (1), both observations recorded during VES and taxa identified from MOTUs for eDNA, while absence (0) when there were neither MOTUs nor observations. In the same way, the confidence interval was calculated using PAMs and RStudio software. R script, PAMs, and occupancy and detection probability data are available at https://doi.org/10.6084/m9.figshare.c.6321395.v1.

A Sankey diagram was created to visualize the species and families detected with eDNA and VES in R.4.1.2 (*R Core Team, 2021*) with the networkD3 package (https://www.r-graph-gallery.com/sankey-diagram.html).

## RESULTS

### Visual encounter surveys

Thirty-six samplings (18 in ponds and 18 in river transects) were carried out for biomonitoring by VES and eDNA methods. VES recorded a total of 248 specimens corresponding to 20 species. One hundred ninety-five specimens were reported in the ponds, and 53 were found in river transects. The most abundant species were: *Boana punctata* (38) and *Dendropsophus bifurcus* (36) at ponds, *Rhinella marina* (18), and *Scinax ruber* (15) at river transects. During the VES, the following species were observed only once: *Allobates* aff. *insperatus*, *Cochranella resplendens*, *Scinax cruentomma*, *Pristimantis variabilis*, and *Adenomera andreae* (Table S5, Supplemental Material).

### 16S rRNA gene reference database

The local reference database of amphibian DNA was developed with 522 16S rRNA gene sequences recorded in Ecuador (https://bioweb.bio/faunaweb/amphibiaweb/), of which 20 sequences of different species resulted from this study (Table S3), 25 sequences were generated in the project "Genetic and ecological characterization of the herpetofauna in an altitudinal transect in the Napo River Basin, and the Colonso Chalupas Reserve, Ecuador" of the Ikiam and 477 were extracted from the NCBI database. In addition, sequences of the most common vertebrates inhabiting the sampling sites, such as fish (five), mammals (five), birds (five), and reptiles (three), were included. This was done to cluster and sort the MOTUs obtained. Bacterial sequences (two) previously sequenced with ONT were also added and used to separate sequences with a higher error rate, as these sequences tended to align with those (bacterial sequences). This was remarked on during analyses and taxonomic assignments. In total, the reference database had 542 sequences from several groups.

## Environmental DNA

The nanopore sequencing run generated a total of 2,704,000 reads. After analysis (Q > 8/Q > 12) with the MinKNOW v21.06.0 and Amplicon_sorter algorithms, we obtained 2,508,000 and 1,204,938 reads, respectively. A total of 375,056 reads were found to belong to amphibians (326,630 for ponds and 48,426 for river transects). The remaining readings corresponded to several groups: fish (479,972), mammals (121,731), birds (2,249), and other groups (225,930). At the end of the analysis, the transport, amplification, and sequencing controls (the two sequenced controls had few reads and none aligned with the vertebrate group) were negative. Table S6 details the list of species and reads.

## eDNA *vs.* VES comparison

Using both eDNA and VES, a total of 33 amphibian species were detected. Individually, eDNA detected 28 and VES 20 species, respectively (13 with eDNA only, five with VES only, and 15 with both methods). Six amphibian families were detected, with Hylidae being the richest with 14 species (three eDNA, one VES, and 10 with both methods), followed by Strabomantidae with nine species (six eDNA, one VES, and two with both methods). Both methods detected all families except for Aromobatidae, which had only a single record (*Allobates* aff. *insperatus*) with VES and was not detected by eDNA (Fig. 3; Table S7). Four species detected only with eDNA are of particular interest: *Pristimantis acerus, P. eriphus, P. mallii,* and *P. sp. (INABIO 15591)*. These species inhabit significantly higher elevations than the study sites, having been recorded at the exploration site within Colonso-Chalupas Reserve (0°56′07.0°S;77°55′36.6°W) at 1,518 m.a.s.l.

eDNA detection probability varied widely between species. The lowest recorded probability was 0.29 (*Oreobates quixensis*), and the highest recorded probability was 1 (both *Rhinella marina* and *Scinax ruber*). This difference may be because *O. quixensis* was detected at only one site (TrP2), while *R. marina* and *S. ruber* were recorded at all sampling sites. The DP for VES had a range of variation, being the minimum for the species *O. quixensis* and *Pristimantis malkini* (DP = 0.056), which were observed twice at the same sampling site (TrP2). Species with a DP > 0.70 were *S. ruber* (DP = 0.71) and *R. marina* (DP = 0.79), detected in different aquatic environments during the six months of monitoring (ChP1, ChP2, TrP1, TrP2, TrP3). The species *Boana geographica* (P = 0.39), *Boana punctata* (P = 0.49), and *Scinax garbei* (P = 0.58) were consistently observed at the same site (ChP2) during the entire sampling time using VES. The DP of these three species when using eDNA corresponds to DP = 0.83 (*B. geographica*), DP = 0.75 (*S. garbei*), and DP = 0.58 (*B. punctata*). This increase in DP with eDNA may be since MOTUs of these species were detected at several sites (ChP3, ChP2, TrP3, TrP2). Including all species, the DP for eDNA was 0.42 CI [0.40–0.45] *vs.* 0.17 CI [0.14–0.20] for VES (Fig. 4; eDNA and VES records, occupancy probability and detection data are available in https://doi.org/10.6084/m9.figshare.c.6321395.v1). The occupancy model was used to help estimate the true occupancy and DP of species at the site with margins of uncertainty (*MacKenzie et al., 2017*).

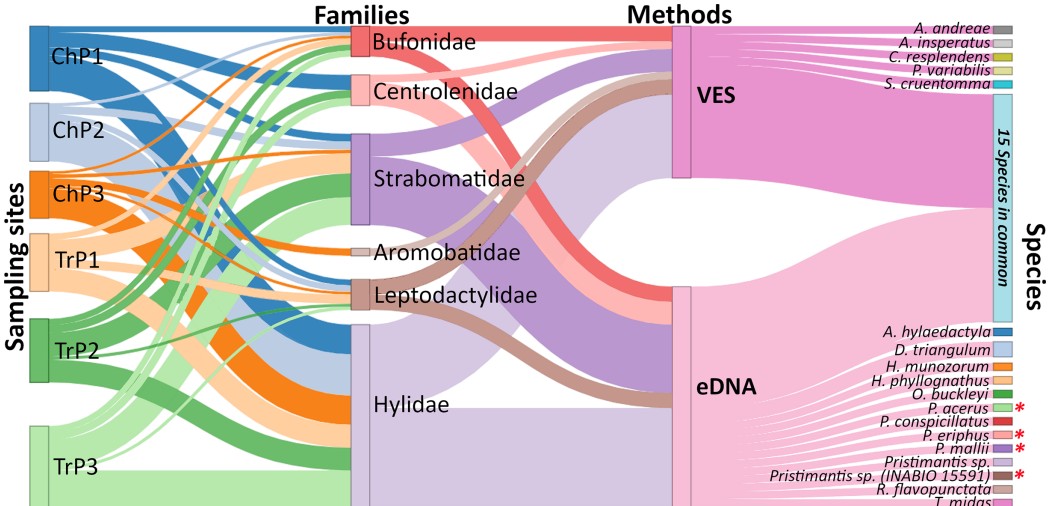

**Figure 3 Relative frequency of amphibian species detection using environmental DNA (eDNA) and visual encounter surveys (VES).** Biomonitoring was for six months in the Tena River micro-basin, Napo, Ecuador. We detail the sampling sites (Ch = ponds, Tr = river transects), families identified, methods applied, and amphibian species detected by each technique. Species with an asterisk (*) are mountain frogs. The thickness of the connecting lines in the diagram corresponds to the number of species identified except for the lines between site-family nodes which corresponds to the ratio (Number of species detected for a family by eDNA and VES/total species record of a family in all sampling) × (number of species detected of the family a site).

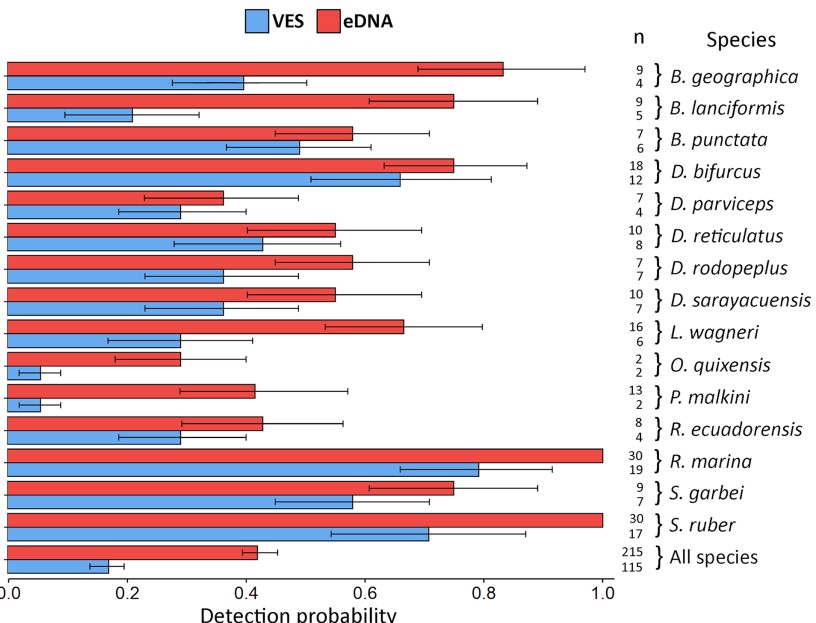

**Figure 4 Detection probability using VES and eDNA monitoring methods.** The probability was calculated for each species detected by the two techniques and for all species combined. Error bars represent the confidence interval (CI) of the analysis and *n* refers to the detection number of each method, for each species and for all species.

## DISCUSSION

The information generated from monitoring organisms using traditional and eDNA-based techniques is essential for ecological and conservation research (*Bálint et al., 2018*; *Angulo et al., 2006*). In this study, we monitored amphibian communities in aquatic environments using VES and eDNA techniques. A reference database that included 20 new DNA barcode sequences generated in this work was constructed. It is essential to mention that the species *Rhaebo ecuadorensis* was not found in the NCBI database due to a labeling error. This was evidenced when the alignment of our sequence with the database was performed, obtaining 99.9% pairwise identity with *R. glaberrimus*, a species from which *R. ecuadorensis* was separated (*Mueses-Cisneros, Cisneros-Heredia & Mcdiarmid, 2012*; *Pramuk, 2006*). It demonstrates that not all genetic information in the NCBI is correctly curated (*Pentinsaari et al., 2020*). For this reason, curating sequences that enter our databases allows us to resolve these discordances.

To perform successful amphibian eDNA biomonitoring in high-biodiversity areas and to recover the true taxonomic composition of environmental samples, it is crucial to have a curated, high-quality reference database (*Taberlet et al., 2018*; *Pawlowski et al., 2021*; *Weigand et al., 2019*). Several scientific reports detail the unfavorable effect on analyses when a database has curation errors, leading to omissions in identifications and generating the discard of sequences that would be considered false positives (*Bohmann et al., 2014*; *Taberlet et al., 2018*; *Schenekar et al., 2020*). Our database was designed for amphibian detection, and 96.3% of the sequences belong to amphibians. For TRMB, this work provides reference sequences for 20 amphibian species, including *A.* aff. *insperatus* and *C. resplendens*, documented for the first time for this area. As mentioned by *Angulo et al. (2006)*, new contributions in terms of species distribution are relevant, especially in unexplored areas, because complete biodiversity inventories help to inform and develop conservation strategies.

### eDNA *vs.* VES

The simultaneous application of eDNA-based and traditional methods in areas with high biodiversity has been analyzed (*Sasso et al., 2017*; *Bálint et al., 2018*; *Herder et al., 2014*) and recommended (*Keck et al., 2022*) in several studies, as these two approaches are compatible when assessing the diversity of organisms. Additionally, using multiple techniques during the biomonitoring of organisms can result in more complete and detailed information compared to monitoring by a single method, as each approach has its characteristics and an established methodology for species detection (*Lopes et al., 2017*; *Angulo et al., 2006*; *Herder et al., 2014*).

In the biomonitoring of amphibians in TRMB, we used eDNA and VES. Combining these two techniques recorded 33 species, 53% of the species previously documented in our study area. eDNA detected 28 species (DP = 42%), while VES recorded 20 species (DP = 17%). These results show that using eDNA can detect more amphibian species compared to VES, in concordance with the meta-analysis performed by *Fediajevaite et al. (2021)*, who compared 49 studies of both techniques.
Although the eDNA-based monitoring technique can detect more taxa than VES, some species can be recorded by both approaches; this will depend on their abundance and distribution in aquatic environments (*Herder et al., 2014*; *Keck et al., 2022*; *Polanco Fernández et al., 2021*). We have detected 15 amphibian species using the combination of both techniques in agreement with the species observed in the buffer zone of the Colonso Chalupas Biological Reserve (*Coloma & Tapia, 2015*). Similarly, these species have been frequently associated with natural lentic water bodies (*Ron, Merino & Ortiz, 2019*; *Lacoursière-Roussel et al., 2016*), where a higher DNA concentration and abundance of anurans would be expected (*Baetens, 2019*; *Petitot et al., 2014*).

On the other hand, several studies mention the presence of a fraction of species that can only be detected by eDNA-based techniques or traditional methods (*Lopes et al., 2017*; *Fediajevaite et al., 2021*). In our case, five species were recorded only by VES, including *A. andreae, A.* aff. *insperatus* and *P. variabilis*, which were detected in previous field sampling in the TRMB (*Ordóñez, Valle & Veintimilla, 2011*; *Coloma & Tapia, 2015*). As *Ron, Merino & Ortiz (2019)* mentioned, these species are terrestrial and diurnal and are usually observed perching in leaf litter or under fallen trees. As such, we would expect water samples to contain low aquatic eDNA concentrations of these terrestrial species, leading to minimal or no detection with aquatic eDNA monitoring, as the inclusion criterion in the sample-organism relationship would not be met (*Taberlet et al., 2018*; *Bruce et al., 2021*). *C. resplendens* and *S. cruentomma* have not been recorded in scientific collections in our study area or neighboring areas but have been recorded in the Napo province (*Duellman, 1972*; *Guayasamin et al., 2006*). These species are associated with water bodies but are rare in disturbed areas, such as our sampling area (*Ron, Merino & Ortiz, 2019*). The record of *C. resplendens* is unexpected mainly because it is considered a species rarely observed in disturbed Amazonian ecosystems. The fact that these five species were not detected with eDNA could result from deficiencies in the capture of DNA molecules due to the non-homogeneous distribution of eDNA in aquatic ecosystems.

Additionally, 13 species were only detected by eDNA, previously recorded in the Napo province and the TRMB (*Heyer et al., 2001*; *Ordóñez, Valle & Veintimilla, 2011*; *Coloma & Tapia, 2015*). However, four of them: *Pristimantis* sp. (INABIO 15591), *P. acerus, P. eriphus*, and *P. mallii* are characterized as mountain frogs that were recorded in the exploration area within the Colonso Chalupas Reserve (Fig. 1). In the study area, these species were detected in different river transects and sampling months. *P. mallii and P. acerus* were identified in January 2020 in TrP1 and March and April 2021 in TrP3. Concerning *P. eriphus* and *Pristimantis* sp. (INABIO 15591) were recorded in March and April 2021 on TrP3 and TrP2.

The presence of eDNA of these four species in the sampling sites may be the result of several factors: geomorphological characteristics of the TRMB, with its wide drainage area of approximately 134.86 km$^2$, an average slope of 22.4%, and an annual rainfall between 3,500–4,000 mm (*Hurtado-Pidal et al., 2020*). All these characteristics could promote the movement of genetic material from species present in the mountainous regions of the upper micro-basin towards the river and downstream to the study area. This statement is corroborated by the displacement of organic material, including DNA, in rivers

(*Pozo et al., 2009*; *Deiner et al., 2016*). Likewise, it is necessary to consider that when working with eDNA in a river transect, it will not have eDNA exclusive to the species of the site. However, there is a probability of detecting species from nearby regions (*Shogren et al., 2017*; *Nevers et al., 2020*).

The MOTUs of these frogs might also be related to cross-contamination during field sampling or laboratory processes. During fieldwork, samples may become contaminated because it is more difficult to preserve sterility. Therefore, using negative transport controls during sample collection and handling allows for knowing at an early stage whether contamination is present or not (*Lopes et al., 2017*; *Baetens, 2019*; *Herder et al., 2014*). Disinfecting and sterilizing all materials and work areas is essential to reduce these problems. There might also be a risk of cross-contamination between samples during library preparation, as different samples are processed simultaneously (*Borst, Box & Fluit, 2004*). This could trigger index swapping, the faulty assignment of sample index. This exchange could also occur in the sequencer when different samples are sequenced together (*Zavala et al., 2022*). Likewise, sequencing errors could generate these MOTUs, especially if the technology's accuracy (ONT) is 99.3%. To reduce these errors and obtain reliable MOTUs, it is necessary to clean up low-quality reads and generate robust consensus sequences from multiple copies of a specific region (*Vierstraete & Braeckman, 2022*; *Baloğlu et al., 2021*).

The negative controls used in each experimental phase of this study were negative, making it possible to verify that mountain frog MOTUs do not derive from cross-contamination or sequencing errors but originate from organic matter (eDNA) outside the sampling area. According to *Bohmann et al. (2014)*, a false positive is when the eDNA detects a species not present at the site. The four mountain species recorded in the lower part of the TRMB would then be considered as false positives for the sites studied. Identifying false positives during the taxonomic assignment is essential to avoid generating biased data that cannot be corroborated in future studies. These errors can be distinguished if a taxonomically curated reference database is available, with species whose biogeographic distributions are appropriate and correspond to the region or area of study.

## Benefits and limitations of eDNA-based and VES techniques

eDNA-based monitoring is a promising and highly efficient technique for biodiversity monitoring, especially in areas where access is complex, and species richness is high, which would require a significant VES sampling effort. Also, direct contact with the organism of interest is not required since environmental samples (water) are used as a source of information. Other advantages of this technique are that it is non-invasive, can be applied in hard-to-reach areas, and reduces monitoring time. However, as the technique is based on DNA barcode sequences, we have limitations in estimating the abundance of organisms, as well as other aspects of their biology (*e.g.*, age, sex), so it is crucial to complement monitoring with scientific collections from VES (*Bálint et al., 2017*; *Ruppert, Kline & Rahman, 2019*).

Taxonomic assignment of MOTUs is limited to a local reference database and relies indirectly on information generated by traditional methods (*Taberlet et al., 2018*;
*Bruce et al., 2021*). As reported in this study, it is necessary to compare all MOTUs to identify them, and biogeographic and ecological information on the detected species is required to discriminate species outside our area of interest, *i.e.*, to determine false positives (*Bohmann et al., 2014*; *Taberlet et al., 2018*; *Shogren et al., 2017*). This is demonstrated by identifying mountain frogs that live from 1,500 m.a.s.l. in sites with altitudes below 600 m.a.s.l. To overcome this limitation in eDNA, several studies propose as an alternative the use of a taxonomy-free approach, which employs a molecular index calculated directly from eDNA data without any reference (*Apothéloz-Perret-Gentil et al., 2017*; *Apothéloz-Perret-Gentil et al., 2021*). This approach could improve taxonomic resolution, which is problematic in the current study, given that not all sequences aligned with a percentage of identity >97%.

Replicating environmental conditions *in vitro* is a real challenge for eDNA-based techniques. Field conditions differ significantly from the laboratory because eDNA is exposed to biotic and abiotic factors, habitat, flow variation, and more (*Valentini et al., 2016*; *Herder et al., 2014*). On the other hand, the standardization of field and laboratory eDNA protocols is essential for the reliable interpretation of results (*Roh et al., 2006*; *Harper et al., 2019*). Similarly, using positive and negative controls is essential for controlled experimentation. A positive control is a sample or reagent used to test for possible PCR inhibition or for DNA degradation during the handlings. In contrast, a negative control is a sample using the same experimental conditions except for the treatment (*Barata, Griffiths & Ridout, 2017*; *Takahashi et al., 2020*) and aims to document the absence of contamination. These protocols follow strict sterilization standards, decreasing contamination risks (*Taberlet et al., 2018*; *Herder et al., 2014*).

eDNA-based monitoring is an ecological diagnostic method that has revolutionized diversity monitoring in different ecosystems (*Taberlet et al., 2018*; *Harper et al., 2019*; *Keck et al., 2022*; *Ruppert, Kline & Rahman, 2019*). Several scientific publications have documented its development and compatibility with other methods, such as VES. They have also demonstrated their sensitivity and efficiency in detecting multiple species in different environments worldwide (*Valentini et al., 2016*; *Bálint et al., 2018*; *Zaiko et al., 2018*). Generating a plan for constantly monitoring amphibians and other groups of interest in the TRMB using VES and eDNA will be fundamental to completing the inventory of the fauna inhabiting this area. This would facilitate the description of new species presumed to be present in unexplored areas of this basin. A complete inventory would provide a clear picture of the population dynamics of endangered species, invasive species, and species with antimicrobial peptides of biomedical interest (*Willette et al., 2021*; *Ficetola et al., 2010*). In short, eDNA and VES combined can generate essential information for developing short- and long-term conservation strategies.

## CONCLUSIONS

This study demonstrates that the eDNA-method had a greater capacity to detect amphibians in Amazonian aquatic environments than VES. However, it highlights the complementary information of traditional methods to complete a database and corroborate the results.

Sixty-two species have previously been recorded in the TRMB, of which 33 were detected using eDNA and VES in this study. This represents a small fraction of the species presumed to inhabit this region, as there are several unexplored areas where the batrachofauna diversity is unknown. Developing a monitoring program using two or more techniques is necessary to generate a complete picture of amphibian diversity in the TRMB. Also, to increase the detection rate of amphibians, using various types of environmental samples could expand the range of detection of terrestrial species or inhabitants of specific micro-ecosystems (*e.g.*, bromeliads, tree holes, leaf litter, *etc.*).

A traditional monitoring approach (VES) has been used for decades and has generated valuable information on biodiversity in some ecosystems, but specific taxonomic knowledge is needed, and sampling is invasive, complex, and time-consuming. eDNA-based techniques, when combined with VES, can generate robust and verifiable records that contribute to the development of conservation strategies in biodiversity hotspots.

## ACKNOWLEDGEMENTS

The authors would especially like to thank all the people who participated in the fieldwork: María José Sánchez, Alex Arias, Álvaro Pérez, Keyko Loza, Karina Quizhpi, Yulissa Ugsha, Jomira Yánez, Viviana Cabrera, Miguel Gómez, Carlos Carrasco, Samanta Oña; in the laboratory processes: Mónica Sánchez; in the bioinformatics analysis: Alex Sánchez, Zane Libke and the map design: Julieth Chancay and Elizabeth Naranjo. We would also like to thank two anonymous reviewers for their valuable comments.

### Funding

This work was funded by VLIR-UOS through the project "DNA-based monitoring for assessing the effect of invasive species on aquatic communities in the Amazon basin of Ecuador" (ECU-SIN277). The funders had no role in study design, data collection and analysis, decision to publish, or preparation of the manuscript.

### Grant Disclosures

The following grant information was disclosed by the authors:
VLIR-UOS: ECU-SIN277.

### Competing Interests

The authors declare that they have no competing interests.

### Author Contributions

- Walter Quilumbaquin conceived and designed the experiments, performed the experiments, analyzed the data, prepared figures and/or tables, authored or reviewed drafts of the article, and approved the final draft.
- Andrea Carrera-Gonzalez conceived and designed the experiments, performed the experiments, authored or reviewed drafts of the article, and approved the final draft.
- Christine Van der heyden conceived and designed the experiments, performed the experiments, authored or reviewed drafts of the article, and approved the final draft.
- H Mauricio Ortega-Andrade conceived and designed the experiments, performed the experiments, authored or reviewed drafts of the article, and approved the final draft.

### Animal Ethics

The following information was supplied relating to ethical approvals (*i.e.*, approving body and any reference numbers):

This research project was carried out through the Framework Agreement for Access to Genetic Resources No. MAATE-DBI-CM-2021-0177, within the project entitled "Detection of key species of fauna (threatened, invasive) and microorganisms (pathogens and microbiota) associated with aquatic ecosystems of Ecuador, with molecular techniques of environmental DNA and metabarcoding".

### Field Study Permissions

The following information was supplied relating to field study approvals (*i.e.*, approving body and any reference numbers):

Ministerio del Ambiente, agua y Transición Ecologica del Ecuador.

### DNA Deposition

The following information was supplied regarding the deposition of DNA sequences:

The 16S ribosomal RNA sequences are available at GenBank: ON907619 to ON907638.

### Data Availability

The data is available at figshare: Quilumbaquin, Walter (2022): eDNA and VES Supplemental Material. figshare. Collection. https://figshare.com/collections/eDNA_and_VES_supplementary_material/6321395/1.

### Supplemental Information

Supplemental information for this article can be found online at http://dx.doi.org/10.7717/peerj.15455#supplemental-information.

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
