# Peer review of "Environmental DNA and visual encounter surveys for amphibian biomonitoring in aquatic environments of the Ecuadorian Amazon"

_PeerJ, doi:10.7717/peerj.15455_

## Round 0.1 · original submission · Major Revisions

I have read the paper and both reviews. I agree with the reviewers in that the paper is interesting but needs major work before it is ready to be accepted for publication. Please read the reviews carefully and modify the text accordingly.

Reviewer 1 ·

Basic reporting

This manuscript mostly meets the basic reporting requirements of PeerJ and eventually will be suitable for publication. However, because there are considerable grammatical and stylistic errors, the manuscript is not acceptable for publication in its current form. The paper has too many citations (100) and could be shortened by as much as 15-20%. I recommend the authors seek out a professional editor to look over the manuscript to improve the readability and make it more concise. Otherwise, this work is self-contained, represents an appropriate publishable unit, and there is sufficient background and context provided.

Experimental design

The article is within the Aims and Scope of PeerJ and the research question is well defined. A clear statement of how this work fills a knowledge gap is presented. The experimental design is well planned and executed. The investigation appears to have been performed to a high standard. The methods are described in sufficient detail and information for the most part, though I would like to know how long samples were stored at -20 prior to eDNA isolation.

Validity of the findings

The metadata are provided on Figshare and the sequences provided in supplementary material. Positive and negative controls were used, an important consideration for eDNA studies. The findings are clearly stated and linked to the original research questions. I found it remarkable that the high-elevation, terrestrial Pristimantis species were detected in the lowland river sites and thought the authors could have highlighted this finding more to demonstrate the utility of eDNA in detecting species present far from the sampling site.

Reviewer 2 ·

Basic reporting

Quilumbaquin et al. conducted repeated VES and eDNA surveys at six sites in the Ecuadorian Amazon and compared the efficacy of the two methods for detecting amphibians. I commend the authors for their approach and think that this is a wonderful application of new eDNA technologies. Exciting research! I think that there is a lot of value in this manuscript and look forward to seeing it eventually published; however, I also think that there are some substantive revisions that should be made to improve the clarity and rigor of the study.

I've provided some more detailed line-by-line comments in "additional comments" below, but I'll try to summarize some major points where relevant in each of these sections, too.

- The English writing is strong throughout the manuscript. (I wish I could write so well in Spanish.). There are places where I would recommend that the authors make some small changes, and I noted some of them in my more detailed comments. If the authors think it would be worthwhile, I'd like to point out the free manuscript review service offered by SSAR (https://ssarherps.org/publications/manuscript-review-service/). This can be really helpful! Again, I found the English to be strong in this manuscript, but I just wanted to point out this resource in case the authors would like another English speaker to review it carefully.

- The references included are mostly appropriate, but there are some places where this should be revisited (e.g., references to metabarcoding in articles without metabarcoding).

Experimental design

I think that the two things that would most improve this manuscript are:

- A more thorough description of the methods (and perhaps, the use of additional methods) for classifying eDNA amplicon data. More detail needs to be provided re: how the reference database samples were curated, how the authors assigned taxonomy to the OTUs they called, and where uncertainty might lie. This would also be greatly improved by sharing the raw sequence data and code used in this study.

- A more thorough descriptions of the methods and results of the occupancy modeling. As written, it is difficult to interpret the results, and providing these details would allow readers to understand the model construction and parameter estimates.

Validity of the findings

No additional comment.

Additional comments

My more detailed line-by-line comments are below:

Line 49: I would recommend changing "allows to generate" to "allows scientists to generate".

Line 52: Some scientists like to distinguish between inventory (i.e., determining which species are present at a moment in time) and monitoring (i.e., evaluating how presence/absence changes over time). Because this study is more similar to the former description, I would recommend changing "monitoring" here to "inventory".

Line 57: I recognize that "ADN" is the standard abbreviation in Spanish, but because the authors use "DNA" throughout the rest of the manuscript, I think I would recommend changing this to "DNA". (I have always liked the sound of "ADN" better, though!)

Line 63: Change "This" to "These".

Line 63: Change "being Hylidae" to "with Hylidae being".

Line 64: I find this presentation of the numbers to be very confusing. For example, how could the authors have detected 33 species total if they detected 24 hylids and 12 strabomantids (i.e., 36 species)? And if the authors detected 24 hylids---with 13 via eDNA and 11 via VES---does that mean that there was no overlap in the list of species detected with these methods (i.e., no species found with both eDNA and VES)? I suspect that the authors must be double-counting somewhere here, and this should be presented in a way that is easier to follow.

Line 66: The copy editor can chime in here, but I think perhaps that "aff." should not be italicized.

Lines 69-70: I would recommend removing this sentence, as there's not enough context here to interpret this statement. Perhaps it could be explained in the Discussion in the main text, but it doesn't need to be in the Results summary.

Line 76: I think that "high degree of detection" is an imprecise and confusing phrase here.

Line 93: Perhaps change "existing" to "traditional", since eDNA has "existed" and been used in vertebrates for more than a decade now.

Line 114: The authors previously discussed eDNA more generally, but this is the first mention of metabarcoding here in the main text. It should be defined, since it is only a subset of eDNA methods. Not all of the citations here are for eDNA metabarcoding (e.g., citation 24 involves PCR amplification with species-specific primers).

Line 116: If the authors want to state that eDNA was first used in 2008, then the earlier definition ("cells and traces of free DNA present in the environment for the detection and monitoring of various organisms") should be revised to indicate that the term "eDNA" only applies to vertebrates (or to metazoans). Because under that definition, the detection of microorganisms (e.g., bacteria) from DNA in environmental samples was first done much earlier than 2008. Furthermore, I don't believe that Ficetola et al. (2008) used eDNA *metabarcoding*, but rather PCR amplification using species-specific primers.

119: What defines "large amounts of DNA"? Is there a citation the authors can provide to indicate that amphibians and fish release a disproportionate amount of DNA?

Line 147: I recommend changing "identified" to "implemented".

Line 148: Change "limites" to "limited".

Line 150: Again here, citation #24 is not an eDNA metabarcoding study. I would strongly encourage the authors to revisit all citations and make sure that they have correctly differentiated between those that use eDNA more generally (e.g., with a PCR assay) and those that are more specifically eDNA metabarcoding studies.

Line 172: If I understand correctly, some sampling was done in ponds (and not just in the river itself). So perhaps this could be reworded (e.g., change "in one section of the Tena River" to "in and around the Tena River"). As written, it sounds like the sampling was only in the river channel itself.

Lines 184-193: How many people searched during these surveys? In what weather conditions?

Line 186: Does this mean that each site was surveyed six times (i.e., Jan, Feb, and March 2020; Feb., Mar. April 2021)? From the results, this seems right. But it might be good to make that explicit here.

195: Was this eDNA collection done the morning after the nocturnal VES survey? Do the authors think it's possible that they might have contaminated the ponds during surveys before collecting eDNA? I'm not especially concerned about this, but if the authors took precautions to avoid this (e.g., decontaminating field gear between sites), this might provide helpful context.

Lines 199-200: If the authors sterilized bottles and reused them, these details should be provided somewhere. This mention does not provide enough detail (e.g., concentration of the bleach, etc.).

Lines 203-204: Which materials were sterilized with ethanol? Earlier, the authors' description made it sound like the bottles were sterilized with bleach.

Line 204: Ethanol is effective at killing some microorganisms, but I'm not sure that it's very effective at removing DNA in this context. Can the authors provide a link to support the use of ethanol to remove DNA from eDNA sampling gear so that readers can find out more information?

Lines 204-206: Was this bottle also reused between sites? Was it sterilized with bleach or ethanol?

Lines 217: What exactly does it mean for these filters to be "saturated"? That they were clogged and no more water would flow through? How much variation in filtered volume occurred among sampling sites?

Lines 215-219: How were filters handled between the filtering and the storage? With sterilized forceps?

Line 238: As written, this sentence makes it sound like the authors performed new sequencing for all species. But as the paragraph goes on to describe, the authors only did this for a subset. Perhaps this could be rephrased.

Lines 235-241: I'm impressed that the authors were able to compile barcodes for all species in the area! Are there any that were not possible to obtain? Any secretive or rare amphibians for which there weren't data available? If so, this should be stated here.

Line 242: Can the authors provide a more specific citation (i.e., within the book cited here) for the need to have >100 species in a reference database? What should researchers do if their study system only has 10 species in a targeted community? I'm not sure I follow the logic of why a database must have >100 sequences.

Lines 243-244: Does this mean all 16S rRNA sequences with metadata suggesting they were collected from Ecuador? Does this include taxa that occur in Ecuador, but for which the only data originated in another country? I think more detail here is necessary.

Lines 244-246: Which fish, mammals, birds, and bacteria? This bit needs much more description.

Lines 246-249: What parameters were used here, and how did the authors evaluate this? Do these 16S primers designed for vertebrates really amplify all bacteria? I don't think that's true, but it's hard to discern what exactly the authors mean here.

Lines 301-303: Because this is described in detail in the next paragraph, I don't think this sentence is necessary here.

Line 332: Should this say 4C (i.e., normal refrigeration temperature) instead of -4C?

Line 344: I think that much more detail is necessary here re: the taxonomic assignment of eDNA sequences. What was the fate of sequences that matched two different reference database entries at 97% or above? If a sequence matched 96% to an amphibian, but not 97%, isn't it likely that this was an amphibian species not represented in the reference database? Was this same threshold used to assign a sequence broadly to "mammals"? Perhaps most important, sequencing errors occur in all eDNA metabarcoding data (but at especially high frequency in ONT data). How did the authors address this in their analyses? I would tentatively recommend looking into some software (e.g., DADA2) that deals explicitly with this problem.

Lines 346-348: Far more information re: the construction of these models is necessary to interpret the results. For example, did the authors use a single-season occupancy model? Did they fit a different model for each species? For each survey method? Were visits in different months considered to be repeated visits? I think that it's a great idea to fit occupancy models, but it's not possible for readers to understand the results without much more detail. I would strongly recommend including the code used to do this analysis (and other analyses) as supplemental material or in a repository.

Lines 380-382: Why five of each of these, and what does it meant to "polarize" the alignment? How did the authors select these 20 taxa? I think further justification and explanation is needed here.

Lines 390-391: Figure S2 shows no evident band on a gel following a PCR. But did the authors still attempt to sequence this negative control? If not, what does it mean that "at the end of the analysis process...sequencing controls were negative"? To most readers, I think this could imply that the authors sequenced the negative control and found no vertebrate eDNA.

Lines 395-401: See earlier comments (Line 64).

Lines 408-426: It is not possible for readers to interpret these detection probabilities without more information in the Methods re: how the models were constructed. Also, because these are occupancy models and estimate both occupancy and detection probability, the authors should report estimates of both parameters along with the uncertainty surrounding these estimates.

Line 426: It's not clear to me how Figure 2 supports this claim.

Lines 458-491: This section seems extraneous to the goals and results of this study, and I would recommend omitting it for brevity and clarity.

Lines 536-550: I think that the authors should consider and acknowledge other (potentially plausible) explanations for these results (e.g., misidentifications in reference database, contamination of samples in the field or lab, errors like index-hopping, etc.).

Supplement: The Figshare link (doi:10.6084/m9.figshare.20161880) doesn't seem to work for me.

Data deposition: Have the authors included their raw nanopore sequencing data anywhere? It would be great to have this available (e.g., through the NCBI Sequence Read Archive) so that other scientists could replicate the analyses.

---

## Round 0.2 · Minor Revisions

I have read the revised version and I agree that it has improved from the first version. I ask kindly the authors to please seek help with the language so it ends up as a very nicely written paper and the readers can fully appreciate the work.

I fully understand how hard it can be to write in a second language so sometimes it is best to find a professional editor.

Reviewer 1 ·

Basic reporting

I have re-reviewed this manuscript that was originally submitted 6 months ago. This version is improved over the previous one, and I commend the authors for their careful revision. As before, this manuscript mostly meets the basic reporting requirements of PeerJ, is self-contained, represents an appropriate publishable unit, and sufficient background and context is provided. However, the article still suffers from too many writing errors (including several sentence fragments) to be acceptable for publication in its current form. I realize the authors are probably not native English speakers, and I empathize with having to write in another language. Although the authors say they have consulted with someone to help with the revisions, I suggest they seek out a professional editor to ensure that "Clear and unambiguous, professional English [is] used throughout." I have highlighted some on the pdf version that I will upload, but I didn't attempt to rewrite.

Experimental design

The article is within the Aims and Scope of PeerJ and the research question is well defined. A clear statement of how this work fills a knowledge gap is presented. The experimental design is well planned and executed. The investigation appears to have been performed to a high standard. The methods are described in sufficient detail to be able to replicate.

Validity of the findings

The findings are clearly stated and linked to the original research questions. I still find it remarkable that the high-elevation, terrestrial Pristimantis species located 13 km from the sampling site were detected in the lowland river samples. I appreciate that the authors acknowledged that cross contamination was a possibility, but that negative controls ruled this out. I still find it hard to believe but do accept the validity of the findings.

Annotated reviews are not available for download in order to protect the identity of reviewers who chose to remain anonymous.

---

## Round 0.3 · accepted · Accept

I have read the revised version of this manuscript and I believe it is now ready to be accepted for publication.